# Synthetic Biology Tools for Engineering *Aspergillus oryzae*

**DOI:** 10.3390/jof10010034

**Published:** 2024-01-03

**Authors:** Hui Yang, Chaonan Song, Chengwei Liu, Pengchao Wang

**Affiliations:** 1School of Life Science, Northeast Forestry University, Harbin 150040, China; 2Key Laboratory for Enzyme and Enzyme-Like Material Engineering of Heilongjiang, College of Life Science, Northeast Forestry University, Harbin 150040, China

**Keywords:** *Aspergillus oryzae*, DNA assembly, synthetic biology, genome editing

## Abstract

For more than a thousand years, *Aspergillus oryzae* has been used in traditional culinary industries, including for food fermentation, brewing, and flavoring. In recent years, *A. oryzae* has been extensively used in deciphering the pathways of natural product synthesis and value-added compound bioproduction. Moreover, it is increasingly being used in modern biotechnology industries, such as for the production of enzymes and recombinant proteins. The investigation of *A. oryzae* has been significantly accelerated through the successive application of a diverse array of synthetic biology techniques and methodologies. In this review, the advancements in biological tools for the synthesis of *A. oryzae,* including DNA assembly technologies, gene expression regulatory elements, and genome editing systems, are discussed. Additionally, the challenges associated with the heterologous expression of *A. oryzae* are addressed.

## 1. Introduction

*Aspergillus oryzae* belongs to the *Ascomycota* phylum, *Eurotiomycetes* class, *Eurotiales* order, *Aspergillaceae* family, and *Aspergillus* genus. It has been widely used for millennia in the fermentation and food processing industries for applications such as brewing soy sauce, kojic acid production, and brewing. Its historical application has underscored its safety [1]. It has been classified as generally recognized as safe (GRAS) by the Food and Drug Administration (FDA) and the World Health Organization [2]. With the genomes of the *A. oryzae* RIB40 strain and the *A. oryzae* 3.042 strain being deciphered, coupled with rapid developments in genetic engineering technology, *A. oryzae* became an ideal host for natural product pathway analysis and value-added compound bioproduction [3,4]. Capitalizing on a robust capacity for protein synthesis, secretion, and posttranslational modification, *A. oryzae* has also shown potential for heterologous protein expression. Recently, the integration of heterologous pathways into the *A. oryzae* genome through precise genetic manipulation has yielded a diverse array of enzymes and secondary metabolites, exemplified by ergothioneine [5], orsellinic acid [6], chevalone E [7], abscisic acid [8], aphidicolin [9], and others. As genetic manipulation techniques continue to evolve, the utilization of *A. oryzae* as a host for heterologous protein expression is increasingly maturing. In this review, important synthetic biology tools developed to date for *A. oryzae*, including DNA assembly, gene expression regulatory elements, genome editing, and *A. oryzae* transformation, are discussed, as along with strategies to increase protein production in *A. oryzae*.

## 2. DNA Assembly Techniques

DNA assembly technology is an enabling technology for synthetic biology research. The demand for precise, versatile, and efficient methods for assembling DNA has become urgent with the development of synthetic biology. In the study of *A. oryzae*, a variety of DNA assembly methods, such as conventional DNA manipulation via restriction/ligation, Gateway, Gibson assembly, and transformation-associated recombination (TAR), are widely used. Here, an overview of the widely employed DNA assembly strategies for *A. oryzae* is presented.

### 2.1. Conventional DNA Manipulation via Restriction/Ligation

Originating in the early 1970s, conventional DNA manipulation via restriction/ligation constitutes one of the pioneering approaches in molecular biology (Figure 1a). This technique entails cleaving the double-stranded DNA target fragment with a restriction endonuclease, yielding single-stranded sticky or blunt ends. Subsequently, the digested fragment is ligated into vector DNA with matching ends, generating a novel vector capable of expressing the target gene. Having existed for more than five decades, this DNA recombination technique has been widely used for *A. oryzae*; for example, Ryuya Fujii et al. successfully harnessed this technique to introduce four heterologous genes into vectors to construct a pathway for aphidicolin in the *A. oryzae* NSAR1 strain [9]. Hiroyuki Uchida et al. used this method to construct a pNAN8142 vector and transfected it into the *A. oryzae* RIB40 *niaD*^−^ strain to synthesize phytate [10].

### 2.2. Gateway

Gateway cloning is a technique for DNA recombination in vitro based on the site-specific recombination system of the λ phage. This technique works by cloning the target DNA sequence between two attL recombination sites in an entry vector to obtain an entry clone. Subsequently, the entry clone is subjected to site-specific recombination with the destination vector containing two attR sites in the presence of the LR Clonase enzyme mixture, which results in the integration of the target DNA sequences into the destination vector in a defined orientation and reading frame, yielding the destination clone (Figure 1b). Pahirulzaman et al. constructed a toolkit for heterologous expression of metabolic pathways in *A. oryzae* using this technology [11]. In this study, two attR sites were inserted between the P*amyB* promoter and the T*amyB* terminator, as was an overexpression cassette comprising *ccdB* and *camR*, allowing integration into the *A. oryzae* genome. As an example of the feasibility of this technology, Cox’s research group successfully introduced all the genes that synthesize strobilurin into *A. oryzae* NSAR1 via the Gateway method and heterologously expressed strobilurin in *Basidiomycetes* [12].

### 2.3. Gibson Assembly

Gibson assembly requires mixing DNA fragments with 15–40 bp overhangs with linearized vectors and adding a mixture of three enzymes, T5 exonuclease, DNA polymerase, and Taq ligase, to obtain the target product (Figure 1c) [13]. The efficiency and convenience of this technique have made it a swift favorite for *A. oryzae* DNA assembly. For example, Mitsunori Fukaya et al. identified the biosynthesis-related gene clusters of chrysophanol and skyrin using Gibson assembly [14]. Similarly, Takusagawa et al. designed an ergothioneine production pathway using the Gibson method in *A. oryzae* [5]. This rapid DNA assembly strategy offers insights into metabolic engineering in fungi so we can better understand the biosynthetic mechanisms of fungal metabolites.

### 2.4. TAR Technology

In the 1980s, Botstein et al. reported that two DNA molecules containing homologous sequences could be homologously recombined in yeast cells [15,16]. Later, a TAR cloning technique based on the free ends of DNA fragments containing homologous sequences capable of efficient homologous recombination in yeast cells was established (Figure 1d) [17]. This technology’s application extends not only to oligonucleotide assembly but also to lengthy DNA fragments and even bacterial genomes. In 2019, Karen E. Lebe et al. used this method to construct a KO plasmid and conducted a combination of directed knockout and heterologous expression experiments to elucidate the genes and biosynthetic steps leading to the construction of the 4,8-dioxabicyclo-[3.2.1]octane motif and oxygenation at C-12 [18]. In 2022, Sukanya Jeennor et al. used this technique to successfully apply a novel pentose-regulated *A. oryzae* promoter (P*xyrA*) to control heterologous gene expression [19].

## 3. Gene Expression Regulatory Elements

Precise gene expression regulation constitutes a cornerstone of synthetic biology endeavors. However, the precise control of gene expression relies on a clear understanding of promoters, open reading frames, and terminators. The construction of metabolic pathways also requires multiple expression cassettes, for which multiple rounds of iterative transformation are often needed. Therefore, screening markers are also critical gene elements for the construction of expression cassettes. Herein, we describe the key elements of promoters, terminators, and screening markers used in *A. oryzae*, concentrating on their roles in controlling gene expression and harnessing heterologous protein expression within *A. oryzae*.

### 3.1. Promoter

The promoter is a key element that controls the level of gene transcription. Constructing metabolic pathways in *A. oryzae* demands the selection of suitable functional promoters, guided by synthetic biology principles. The selection of an appropriate promoter to drive gene expression depends on the type of target product and the associated metabolic flux in the cell. The promoters in *A. oryzae* include constitutive and inducible promoters. A constitutive promoter does not require a specific inducer for expression. Compared with constitutive promoters, inducible promoters are more easily affected by environmental changes. Under the stimulation of specific environmental signals, the expression of genes can increase. Therefore, inducible promoters are more suitable for the expression of genes with growth defects due to overexpression.

Several carbon-dependent inducible promoters, such as the sorbitol-sensitive promoter (P*sor*) [20], alpha-amylase gene promoter (P*amyB*) [21,22], glucoamylase gene promoter (P*glaA*) [23], xylose reductase gene promoter (P*xyrA*) [19], and α-glucosidase gene promoter (P*agdA*) [24], which are all applied in the production and characterization of recombinant proteins, have been characterized. In addition, the inducible promoter P*thiA* is dependent on thiamine regulation. The promoter inhibits the expression of target genes in the presence of thiamine (vitamin B1), and the expression level of these genes can be controlled by the concentration of external thiamine, which can be applied to the expression of genes that cause growth defects during overexpression [25]. In addition to inducible promoters, many constitutive promoters, such as the constitutive promoter P*gpdA* [26] and phosphoglycerate kinase gene promoter (P*pgkA*) [27], which have also been applied to the production of recombinant proteins in *A. oryzae*, have been identified. In addition, the translation-elongation factor 1 alpha gene promoter (P*tef1*) was cloned from *A. oryzae* KBN616 for the expression of the *pgaA* and *pgaB* genes [28].

In addition, a glucose amylase-encoding gene promoter (P*glaB*) [26] suitable for the production of recombinant proteins via solid-state fermentation of *A. oryzae* was developed. The tyrosinase-encoding gene promoter (P*melO*) is completely opposite to that of P*glaB*. It is inhibited in solid-state culture, and its expression efficiency is much greater than that of P*amyB* and P*glaA* under liquid fermentation conditions [29]. Since P*glaB*, a promoter capable of efficiently expressing recombinant proteins under solid-state culture conditions, was induced by maltose and repressed by glucose, the promoter of the hemolysin gene (P*hylA*) was mined to obtain a promoter capable of being glucose independent and was efficiently expressed under solid-state culture conditions [30].

Additionally, modifications to the promoter region can markedly change transcription efficiency. For example, P*enoA*142 was obtained by introducing 12-tandem Regions IIIa and IIIb into the StyI site of P*enoA* (P*enoA* on the pNGEG plasmid), which exhibited more than 20-fold increased activity [31]. Similarly, P*glaA*142, created by incorporating 12-tandem repeat cis-acting elements Region III and the CAAT box into P*glaA*, boasted promoter activity four to six times greater than that of the native promoter [32]. The activity of recombinant tannase produced by P*glaA*142 in *A. oryzae* was two to eight times greater than that in *Pichia pastoris* [33]. The various promoter characterization methods used are shown in Table 1.

### 3.2. Terminator

The terminator, a DNA-sequence-specific element, governs the cessation of transcription within the transcription unit by initiating the detachment of newly synthesized RNA from the transcription machinery. Typically located after the 3′ regulatory sequence of a gene, terminators have garnered attention for their pivotal role in RNA synthesis processes. Research has underscored the influence of terminators on the RNA half-life and, in turn, on gene expression outcomes [35]. Widely used terminators in *A. oryzae* include T*amyB*, T*adh*, T*eno,* and T*gpdA*, which are used in the PTYGS expression vector set to facilitate multigene expression in *A. oryzae* NSAR1 [36]. In addition, terminators such as T*amyA* [37] and T*rpC* [38] from *A. nidulans* showed trans-species applicability in *A. oryzae*. Nonetheless, despite promising implications, in-depth investigations into the terminator functionality of *A. oryzae* have been limited. Future exploration of the role of terminators in the regulation of gene expression within *A. oryzae* is warranted.

### 3.3. Selection Marker

For the genetic manipulation of filamentous fungi, the choice of suitable screening markers and compatible hosts plays a pivotal role. At present, the main screening markers for *A. oryzae* include auxotrophic selectable markers and dominant selectable markers. Dominant selectable markers include the pyrithione resistance gene (*ptrA*) [39], the carbon toxin resistance marker gene (*AosdhB* (*cxr*)) [40], and the bleomycin resistance marker gene (*Blmb*) [41]. In the past two years, Todokoro et al. identified a new pyridine thiamine resistance marker gene (*thil*) from *A. oryzae* by applying UV mutagenesis [42]. *AosdhB* (*cxr*) and *Blmb* require a special drug-resistant host, so they are not commonly used as screening markers for *A. oryzae*. The *ptrA* marker can be used for non-pyrithione-resistant strains and is widely used in the study of *A. oryzae*. While the use of dominant selectable markers is operationally straightforward, it can induce drug resistance and incur high screening costs.

Compared with the use of dominant selectable markers, auxotrophic selectable markers are more widely used in *A. oryzae* due to their versatility. The main auxotrophic selectable markers used in *A. oryzae* include the *pyrG* gene encoding orotate nucleoside-5-phosphate decarboxylase [43], the *_S_C* gene encoding ATP sulfatase [44], the *argB* gene encoding ornithine carbamoyl transferase [45], the *niaD* gene encoding nitrate reductase [46], the *adeA* gene encoding aminoimidazole nucleotide synthase [47], the *amdS* gene encoding acetamide enzyme [48] and the *adeB* gene encoding phosphoribosylaminoimidazole carboxylase [47]. *A. oryzae* hosts harboring multiple selection marker gene deficiencies in advance are needed for simultaneous use of multiple selection markers. For instance, the double auxotroph transformation system (*niaD*^−^, *sC*^−^) developed by Yamada et al. enables two rounds of genetic operation within a single host strain, which further evolves into triple (*niaD*^−^, *sC*^−^, *adeA*^−^/*adeB*^−^) and quadruple (*niaD*^−^, *sC*^−^, *argB*^−^, *adeA*^−^*/adeB*^−^) auxotrophic mutants [44,47,49], laying the groundwork for gene cluster resolution. Table 2 lists the widely used host strains and selection markers in *A. oryzae*.

The number of screening markers available for *A. oryzae* is quite limited. To overcome this problem, a recovery system in which individual screening marker genes can be reused was developed for *A. oryzae.* Maruyama et al. designed this approach to place the marker gene *pyrG* so it was flanked by two adjacent short repeat sequences. When the transformants were screened in the presence 5-FOA (a compound that causes *A. oryzae* death when *pyrG* is expressed), homologous recombination occurred between the two repeat sequences, yielding *A. oryzae* that had lost *pyrG* [50]. In addition, the reuse of marker genes can be achieved through the *Cre*/*loxP* recombinase system [51].

**Table 2 jof-10-00034-t002:** Selectable markers and nutritional sources of *A. oryzae* strains.

Host Strain	Origin	Genotypes	Nutritional Sources	References
Carboxin-resistance mutant	RIB40	*AosdhB*	carboxin	[40]
Bleomycin-resistance mutant	RIB40	*Blmb*	bleomycin	[41]
NS4	RIB40	*niaD*^−^, *_S_C*^−^	NO_2_, methionine	[44]
NSPID1	RIB40	*niaD*^−^, *_S_C*^−^, Δ*pyrG*, Δ*ligD*	NO_2_, methionine, uracil	[50]
niaD300	RIB40	*niaD* ^−^	NO_2_	[52]
AK2	AK	*ade*^−^, *argB*^−^	adenine, arginine	[53]
NSR13/NSR1	RIB40	*niaD*^−^, *_S_C*^−^, *adeA*^−^	NO_2_, methionine, adenine	[49]
NSA1	RIB40	*niaD*^−^, *_S_C*^−^, Δ*argB*	NO_2_, methionine, arginine, citrulline	[49]
M-2-3	RIB40	Δ*argB*	arginine, citrulline	[54]
NSAR1	RIB40	*niaD*^−^, *_S_C*^−^, Δ*argB*, *adeA*^−^	NO_2_, methionine, arginine, citrulline, adenine	[49,54,55,56]
PTR26	HL1034	*ptrA*	pyrithiamine	[39]
3.042/Δ*pyrG*	3.042	Δ*pyrG*	uracil, uridine	[57]

## 4. Methods of Transformation

The establishment of high-efficiency transformation methods is a prerequisite for the industrial application of gene manipulation and recombinant gene expression in *A. oryzae*. Due to the cell wall structure of filamentous fungi, the transformation efficiency of *A. oryzae* is lower than that of yeast and *E. coli*. A variety of transformation methods have been used in *A. oryzae* to increase the efficiency of transformation. Methods such as electroporation and gene guns, despite their potential, are expensive and have very poor transformation rates, thus rendering them unsuitable for genetic transformation of *A. oryzae* [58]. In contrast, (PEG)/CaCl_2_-mediated protoplast transformation (PMT) [59] and *Agrobacterium tumefaciens*-mediated transformation (ATMT) [60,61] have emerged as widely adopted methods for treating *A. oryzae*. PMT initially loads heterologous genes onto auxotrophic markers and dominant selectable marker vectors as described above, followed by the transformation of protoplast cells through high osmotic pressure induced by transformation reagents. The vector carrying the target gene can be integrated into the host chromosome by nonhomologous end joining (NHEJ) or homologous recombination (HR). NHEJ leads to the integration of transformed DNA into nonspecific genomic loci, and the recombination of homologous sequences leads to the integration at specific genomic loci [62]. The PMT method is simple and effective, is easy to carry out, does not require expensive equipment, and can cotransform multiple DNA fragments. The disadvantage of these methods is that protoplast culture is difficult, the regeneration frequency is low, and the reagent requirements are high [63]. Therefore, the prerequisite and basis for transformation using the PMT method is the preparation of highly productive protoplasts.

ATMT utilizes *A. tumefaciens* to insert T-DNA into the fungal host’s genome. The central process of ATMT is *A. tumefaciens* binding to wounded parts of dicotyledonous plants, triggering T-DNA transfer from the Ti plasmid to the plant cell genome. ATMT, which was initially effective in plant transformation, has been utilized in many animals, bacteria, and fungi. In fungi, it was first applied in yeast [64] and subsequently found effective in other filamentous fungi, including *A. fumigatus* [65], *A. terreus* [66], *A. flavus* [67], and *A. niger* [68]. Compared with PMT, ATMT is characterized by simplicity of operation and superior conversion efficiency. In 2016, Nguyen et al. introduced ATMT in *A. oryzae* for the first time using *pyrG* as a screening marker. It was fused with the green fluorescent protein GFP to establish an overexpression system for the successful *Agrobacterium tumefaciens*-mediated transformation of *A. oryzae* [61]. Subsequently, the author successfully established a *pyrG*/*ptrA* double screening system in the *A. oryzae* 3.042 strain by using the ATMT transformation method [60]. The establishment of the *Agrobacterium tumefaciens*-mediated *A. oryzae* transformation system showed that ATMT has considerable potential for inserting T-DNA mutations in filamentous fungi, thus opening up new avenues for successful genetic transformation.

## 5. Genome Editing Techniques

Genome editing is a crucial tool for research on gene function in *A. oryzae*. This technology empowers remarkably efficient targeted gene modifications, allowing the replacement, insertion, or deletion of target genes to realize the desired phenotypes. Currently, zinc finger nucleases (ZFNs), transcription-like activator effector nucleases (TALENs), and clustered regularly interspaced short palindromic repeats/CRISPR-associated nuclease 9 (CRISPR/Cas9) are the three most extensively used enzymes in genome editing. These enzymes initiate a DNA double-strand break (DSB) at the target gene sequence, followed by two repair mechanisms, NHEJ (nonhomologous end-joining) and HR (homologous recombination). NHEJ promptly repairs DSBs by ligating DNA ends, resulting in gene knockouts. Conversely, HR relies on undamaged homologous regions as templates to mend DSBs with precise insertions or deletions, thereby achieving gene knockouts (Figure 2a). TALENs and CRISPR/Cas9 are pivotal gene editing enzymes in *A. oryzae* genetic manipulation research [69].

TALENs consist of transcription activator-like effectors (TALEs) and the FokI nuclease. TALE proteins from *Xanthomonas* bind to target DNA sequences via a recognition pattern involving repetitive variable two-residue RNDs corresponding to 1 base. A TALE fuses with a FokI monomer to form a TALEN (nuclease), while the FokI nuclease can only exert its activity when it forms a dimer. To promote dimerization, two TALEN monomers are usually designed with closely positioned binding sites. The distance between these TALEN target sequences is typically approximately 15 bp (Figure 2b) [70]. To date, TALENs have been designed to modify the genome by means of DNA double-strand breaks (DSBs) in many organisms. In the wild-type strain *A. oryzae* RIB40, Mizutani et al. designed the targeting site of high-efficiency PtFg-TALENs as the *sC* gene encoding ATP sulfonylase and obtained a variety of mutation patterns by means of NHEJ repair, including the deletion of small, medium, and large gene fragments. Remarkably, the author performed genome editing based on PtFg-TALEN on the *A. oryzae ligD* disruptor (Δ*ligD*), which lacks the *ligD* gene involved in the final step of NHEJ repair, and found that the mutation still existed as the wild type. The mutation mode is based mainly on small fragment deletion [71]. In general, the TALEN system is a valuable tool for genome editing and research into the unique DNA repair mechanisms in *A. oryzae*.

The CRISPR/Cas9 system, which originated as a bacterial and archaeal acquired immune mechanism, has transitioned into a potent genome editing system. This system employs RNA-guided recognition of specific DNA sequences to induce cleavage, which has been the pivotal focus of recent research (Figure 2c) [72]. In 2016, the CRISPR/Cas9 system was initially applied to targeted mutagenesis in *A. oryzae* [73]. In this system, recombinant plasmids harboring Cas9 and sgRNA components are typically constructed in filamentous fungi. The codon-optimized Cas9 gene was transcribed using the P*amyB* strong promoter and terminator. Overexpression of Cas9 did not affect the natural growth of the *A. oryzae* strains. SgRNAs were placed between the U6 promoter and the U6 terminator from *A. oryzae* for transcription. Three target genes, *wA*, *pyrG*, and *yA*, were edited. The results showed that 1 bp insertions or 1–21 bp deletions could be generated at editing efficiencies ranging from 10% to 100% through NHEJ repair [73]. In 2019, Maruyama’s team optimized the CRISPR/Cas9 editing system for *A. oryzae* based on the recyclability of the AMA1 autoreplicative plasmid harboring the resistance marker *ptrA* [74]. This optimization significantly enhanced the editing efficiencies of the target genes *wA*, *pyrG*, and *yA*, reaching 55.6% to 100%, and 68.1% for the simultaneous knockdown of *wA* and *niaD*. With HR repair, simultaneous knockout of *wA* and *niaD* was accomplished using a circular donor-DNA plasmid without introducing a marker gene, achieving 61.9% knockout efficiency. At the same time, Liu et al. developed the high-expression sites (hot spots) HS201, HS401, HS601, and HS801. The erinacine biosynthetic gene cluster was reconstructed by knocking in 11 genes using CRISPR/Cas9 technology, and the transformation efficiency was 100% [75].

In addition to plasmid-mediated transformation expressing Cas9 and sgRNA, genome editing in filamentous fungi has also been achieved using the direct introduction of Cas9/sgRNA ribonucleoprotein complexes. Compared with the former, it is characterized by simple manipulation and speedy completion. Using the CRISPR/Cas9 system, Hiroki Ishida et al. succeeded in obtaining a pyridostigmine-resistant mutant by introducing the Cas9/sgRNA ribonucleoprotein complex to target the *thiI* gene. The mutant did not produce any nutritionally deficient phenotype [42]. In addition, the simultaneous introduction of ribonucleoprotein complexes for *thiI* and *wA* or *sreA* produced double mutants under pyrithiamine resistance selection. The mutation frequencies were 5.5% and 8.2%, respectively. Overall, a loss-of-function mutation in the *thiI* gene emerged as a new marker gene for the generation of target gene mutants in *A. oryzae*. Notably, Katherina Garcia Vanegas et al. successfully used Mad7 (ErCas12a, an alternative CRISPR nuclease, Mad7) to introduce nonspecific and specific template-directed mutations in *A. oryzae*, including gene disruption, gene insertion, and gene deletion, thereby expanding the CRISPR toolbox in fungal cell factories [76].

## 6. Strategies for Heterologous Protein Production in *A. oryzae*

Compared with prokaryotic expression systems such as *E. coli*, fungal cell expression systems offer proper posttranslational modifications, protein folding, and protein secretion [77]. As one of the most important strains in the fermentation industry, *A. oryzae* exhibits a strong protein secretion capacity and has received increasing attention. However, the production of heterologous proteins using *A. oryzae* secretion often involves challenges related to transcription, translation, secretory processes, and extracellular degradation [77,78]. These problems make it difficult for *A. oryzae* to produce foreign proteins and thus achieve the desired results. Consequently, optimizing heterologous expression systems is of paramount importance.

### 6.1. Reduction in Autoprotease Activity in A. oryzae

High expression and efficient secretion of heterologous proteins are prerequisites for the production of heterologous proteins, but *A. oryzae* contains a large number of proteolytic enzymes that can degrade heterologous proteins [79]. Thus, proteolytic degradation inhibits the expression of heterologous proteins to some degree. To increase heterologous protein yields, researchers have applied various strategies to reduce protease activity. Jin et al. achieved a 63% increase in heterologous human lysozyme (HLY) production by deleting the *tppA* and *pepE* genes [80]. By knocking down *tppA* and *pepE*, Yoon et al. disrupted 10 protease genes to reduce the protease activity of *A. oryzae* itself, further enhancing the production of human lysozyme (HLY) and the heterologous protein bovine chymosin (CHY) [50,81]. However, in previous studies, HLY expression cassettes carrying the *_S_C* selection marker could not be removed because they were randomly integrated into the genome. Therefore, Nemoto et al. used *A. oryzae* NSAR1 to construct *tppA-* and *pepE*-deficient strains and obtained NS-tApE. The HLY expression cassette carrying the *niaD* selection marker was homologously recombined to the *niaD* locus, after which the HLY mutant was isolated by means of UV mutagenesis. The yield of HLY was 92% greater than that of the parent strain. Then, the HLY expression cassette carrying the *niaD* selection marker was removed from the mutant via positive selection with chlorate medium to obtain the AUT strain. The AUT strain could be used to produce other heterologous proteins. Based on this strain, the HLY expression cassette was reintroduced into the AUT strain to obtain AUT1. It was demonstrated that the mutation caused by UV mutagenesis occurred on the chromosome of the AUT strain rather than in the HLY gene expression cassette [82]. On the basis of AUT1, Nemoto et al. identified a new gene, *autA,* by comparing whole-genome sequences. The mutation or deletion of this gene is related to HLY yield [83]. *A. oryzae* secretes a large amount of Taka-amylase A (TAA) into the medium, which affects the production of heterologous proteins. To solve this problem, Noriyuki et al. performed RNA interference (RNAi) on the *taa* gene of *A. oryzae* KBN616 by knocking out five proteases (*alp*, *npII*, *pepE*, *npI*, and *pepA*). The results showed that the protease activity of *A. oryzae* itself was significantly reduced, and the amylase activity completely disappeared [84].

### 6.2. Optimization of the Secretion of Heterologous Protein Pathways

Although *A. oryzae* has a robust ability to secrete extracellular proteins, issues sometimes arise in heterologous protein production. High expression of heterologous proteins leads to endoplasmic reticulum (ER) overload, resulting in the accumulation of misfolded or unfolded proteins within the ER, a state known as ER stress or protein secretion stress. Only correctly folded, modified, and assembled functional proteins are exported to the Golgi and are not retained or degraded in the ER (Figure 3). Cells resist ER stress through different stress response mechanisms, including the unfolded protein response (UPR) and endoplasmic-reticulum-associated protein degradation (ERAD). Therefore, enhancing heterologous protein production relies on effective secretion pathways and proper protein folding. For example, overexpression of *BipA* and protein disulfide isomerase (*PDI*) can effectively increase heterologous protein production [85]. In the secretory pathway, Yoon et al. disrupted the gene encoding the vacuolar protein sorting (Vps) receptor AoVps10, resulting in reduced transport of recombinant proteins to vacuoles through the CPY pathway and increased secretion of recombinant proteins in the culture medium [86]. Huy-Dung Hoang et al. confirmed that the putative endoplasmic reticulum Golgi cargo receptors AoVip36 and AoEmp47 affect the secretion of heterologous proteins [87]. Autophagy has been reported to transport misfolded secreted proteins accumulated in the ER to vesicles [88]. In contrast, heterologous proteins are usually considered to be misfolded proteins when they are produced, which is detrimental to their production. Yoon et al. destroyed several autophagy genes in *A. oryzae*, and compared with that of the parent strain, the production level of CHY was increased three-fold [89]. In addition, truncation or deletion of the promoter of the amylase A gene (P*amyB*) led to the identification of cis-elements essential for RESS. Cis-elements help alleviate the inhibition of fungal protein expression under secretory stress and relieve endoplasmic reticulum stress. This approach provides an effective way to facilitate the industrial application of cell factories for the production of valuable recombinant enzymes and proteins [90].

### 6.3. Fusion Expression Strategies

To ensure efficient transport and secretion of heterologous proteins, fusion expression of heterologous proteins with well-secreted proteins or the addition of signal peptide sequences from efficiently secreted proteins at the N-terminus has been employed [91]. For instance, a fragment of bovine chymosin (CHY) cDNA was fused to a partial fragment of the *A. oryzae* glucoamylase gene (*glaA*). Compared to the direct use of the P*glaA* promoter, the expression of CHY was dramatically increased [92]. Fusion expression of CHY with the amylase (P*amyB*) gene increased the yield of expressed proteins by approximately two-fold compared to that of nonfusion strains of *A. oryzae* [93].

**Figure 3 jof-10-00034-f003:**
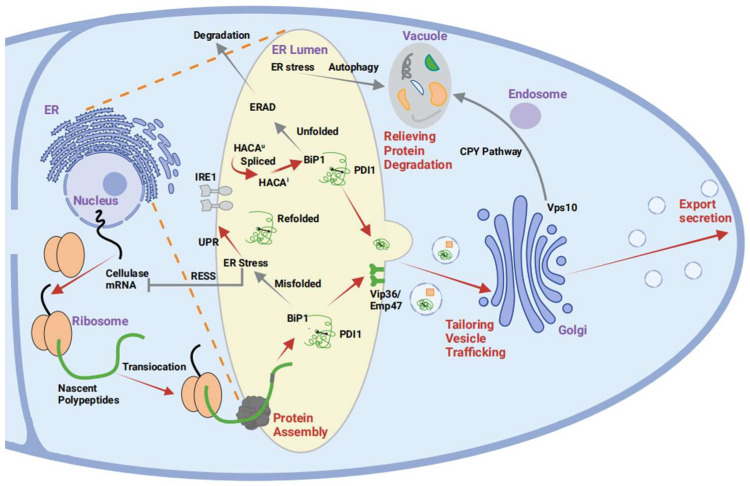
Pathway of heterologous protein secretion by *A. oryzae*. The secretion of foreign proteins begins with protein translation in ribosomes and then complex assembly in the endoplasmic reticulum (ER). The polypeptide entering the ER cavity needs to be folded with the assistance of the binding protein BiP1 and the protein disulfide isomerase PDI1. The correct protein is transported to the Golgi apparatus by vesicles for further modification and packaging. Finally, the protein is transported to the extracellular space through vesicles (red arrows); incompletely folded or misfolded proteins promote protein degradation through the endoplasmic reticulum stress response (gray arrows). In the process of protein degradation, the UPR can trigger the downstream UPR cascade through membrane-bound IRE1 and subsequently couple and autophosphorylate to form activated IRE1. Activated IRE1 matures the mRNA of the transcription factor HAC1 by splicing, leading to the expression of downstream protein-folding genes or degradation by ERAD [94]. The cell structure is represented by purple. The heterologous protein secretion pathway is shown in red font. The yellow dotted line represents an enlarged endoplasmic reticulum lumen. It is worth noting that this figure only shows what we mentioned in this article. The specific process needs to be further explored.

## 7. Conclusions and Perspectives

Efficient genetic modification technology and mature genetic manipulation tools greatly facilitate the development of cell factories. As an important industrial microorganism, *A. oryzae* has attracted widespread attention from researchers due to its complete genome sequence, powerful secretion of proteins, and increasingly mature synthetic biology tools [79,95]. This review has provided an overview of prevalent synthetic biology tools, genetic engineering methodologies, and heterologous expression systems used for *A. oryzae*. However, there are still some issues that need to be addressed.
(1)DNA assembly is one of the limiting factors for the rapid development of a synthetic biology toolkit for *A. oryzae*, overcoming limitations associated with long or multiple fragments and scarred DNA after ligation. Enhanced cloning strategies and DNA assembly techniques are essential for the swift construction of gene expression cassettes. Recently, several biotechnology methods for the genetic modification of filamentous fungi, such as the Modular Cloning system or the Golden Braid-based Fungal Braid system, have been used to assemble strains more quickly in a standardized and modular manner. These methods have potential application value in *A. oryzae* [96,97]. In the future, it is necessary to vigorously develop improved cloning strategies and DNA assembly techniques, to enable faster and more efficient construction of multiple gene expression cassettes.(2)Promoters and terminators are the most important basic elements in synthetic biology research. To date, several gene expression control elements available for *A. oryzae* have been identified. However, compared with the model microorganism *Saccharomyces cerevisiae*, such elements are still rare. It is necessary to construct promoters with shorter sequences, stronger functions, and wider ranges of transcriptional activity to further explore new synthetic biological elements.(3)Gene editing technology has accomplished rapid editing of *A. oryzae* with the advantages of simplicity, high efficiency, and high specificity and accelerated the development of engineered strains. CRISPR/Cas9 systems must address off-target effects and improve the efficiency and enhancement of the efficiency of precisely targeted editing, among other aspects. In addition, CRISPR-related derivative technologies based on dCas9 or nCas9 need to be developed for *A. oryzae*, such as CRISPRa and CRISPRi-mediated gene expression regulation technology and multifunctional CRISPR-mediated combined regulation technology [98,99,100]. The CRISPR system is expected to be rapidly developed and technologically innovated in the future for research on gene function, metabolic pathway reconstruction, precise expression regulation, and high-performance chassis construction in *A. oryzae*.(4)In recent years, research on the production capacity of heterologous protease preparations and other useful substances of *A. oryzae* has greatly improved the production and application of *A. oryzae* [101,102,103]. Future research could focus on the activity of enzyme extraction, the adverse effects of high concentrations of products on *A. oryzae* itself, and the utilization of raw materials. This study provides additional effective methods and research space for the breeding of *A. oryzae* production strains in the future.

In general, *A. oryzae* has become a robust tool for synthetic biology research, but further refinement and optimization of the system are ongoing needs. In the coming years, research efforts must be directed toward providing innovative tools and techniques for engineering *A. oryzae* strains, better maximizing protein expression yields, and facilitating the development and application of natural product synthesis.

## Figures and Tables

**Figure 1 jof-10-00034-f001:**
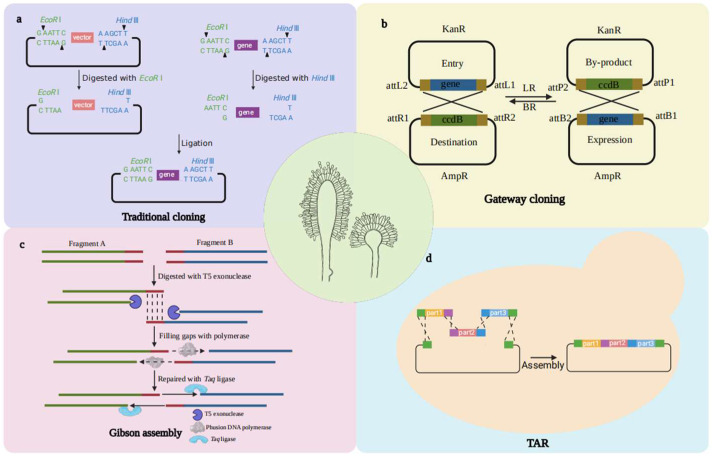
DNA assembly techniques used for *A. oryzae*. *A. oryzae* is indicated by a green module (**a**) Conventional DNA manipulation through the restriction/ligation method. The linear vector and gene fragments are digested with EcoRI and HindIII to obtain fragments and vectors with sticky ends. The gene is successfully assembled into a vector with ligase.The purple module is indicated. (**b**) In the Gateway method, targeted recombination through the att locus enables the transfer of target genes. The yellow module is indicated. (**c**) In Gibson assembly, the 5′ exonuclease activity of the T5 exonuclease is used to cut fragment A and fragment B, and then Phusion DNA polymerase is used to fill the vacancy, and Taq ligase is used to repair the incision. The red module is indicated. (**d**) In TAR, efficient homologous recombination in yeast cells is achieved using the free ends of DNA fragments containing homologous sequences. The blue module is indicated.

**Figure 2 jof-10-00034-f002:**
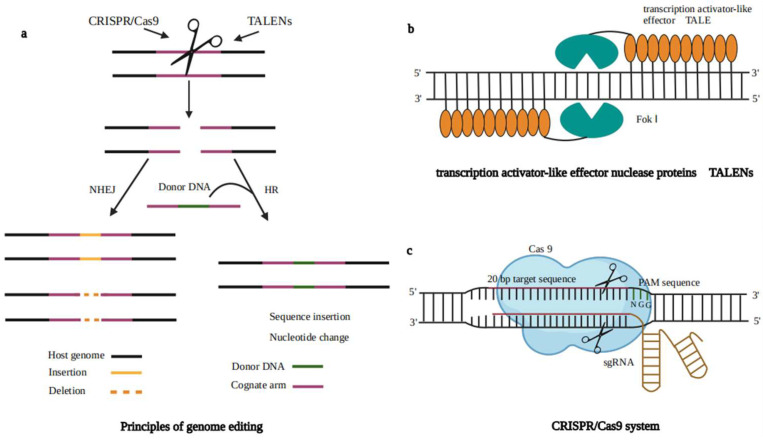
Two repair pathways mediated by the engineered nucleases TALENs and CRISPR/Cas9 in cells. (**a**) In *A. oryzae*, TALENs and CRISPR/Cas9 can recognize and bind to specific DNA sequences, resulting in double-strand breaks (DSBs) that induce NHEJ or HR. NHEJ introduces random insertions and deletions into the genome. HR is the use of the homologous recombination principle to achieve precise integration of exogenous sequences. Notably, Cas9 introduces blunt breaks, while FokI, the TALEN endonuclease, introduces a staggered cut (for simplicity, this difference is not shown in the figure). (**b**) The TALE monomer of TALENs specifically recognizes the DNA sequence and binds to double strands. The fused Fok dimer, composed of two monomers, cleaves the target DNA sequence to produce DSBs. TALE is composed of an N-terminal transport signal, a transcriptional activation domain, a DNA-specific recognition binding domain and a C-terminal nuclear localization signal peptide. FokI: a restriction endonuclease. Shear activity can be exerted only as dimers. (**c**) The CRISPR/Cas9 system containing the Cas9 protein and sgRNA (single-guide RNA). The sgRNA recognizes a 20 bp sequence followed by PAM, after which the Cas9 protein is used to cut the DNA double strand to produce DSBs. PAM sequence: Usually, the PAM sequence is composed of three bases, NGG.

**Table 1 jof-10-00034-t001:** Promoter for *A. oryzae* and its application.

Promoter	Species	Gene Source	Application	References
P*xyrA*	*Aspergillus oryzae*	xylose reductase	For the expression of β-glucuronidase	[19]
P*sor*	*A. oryzae*	sugar transporter-like protein (*stl1*) and sorbitol dehydrogenase (*xyl2*)	For *eGFP* expression	[20]
P*glaA*	*Aspergillus niger*	glucoamylase	For expression of glucosidase	[23]
P*thiA*	*A. oryzae*	thiamine thiazole synthase	For *eGFP* expression	[25]
P*amyB*	*A. oryzae*	alpha-amylase	For expression of lysozyme	[21,22]
P*melO*	*A. oryzae*	tyrosinase	For expression of glucoamylase	[29]
P*SodM*	*A. oryzae*	manganese superoxide dismutase	For expression of cellulase	[29]
P*hylA*	*A. oryzae*	hemolysin-like-protein	For the expression of endo-1,4-β-glucanase	[30]
P*agdA*	*A. oryzae*	α-glucosidase	For the expression of α-glucosidase	[24]
P*pgkA*	*A. oryzae*	phosphoglycerate kinase	For expression of PEP carboxylase	[27]
P*enoA*	*A. oryzae*	enolase	For the expression of β-glucuronic acid	[34]
P*adh*	*Aspergillus nidulans*	alcohol dehydrogenase	For the synthesis of tenellin	[11]
P*enoA*142	*A. oryzae*	improved P*enoA*	For expression of β-Glucuronidase	[31]
P*glaA*142	*A. oryzae*	improved P*glaA*	For the production of tannase	[33]
P*tef1*	*A. oryzae*	translation-elongation factor 1 alpha	For expression of polygalacturonase	[28]
P*glaB*	*A. niger*	glucose amylase	For expression of β-glucuronidase	[26]
P*gpdA*	*A. nidulans*	glyceraldehyde-3-phosphate dehydrogenase	For expression of β-galactosidase	[26]

## Data Availability

All relevant data generated or analyzed during this study are included in this manuscript.

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
