# Peer review of "Synthetic Biology Tools for Engineering Aspergillus oryzae"

_jof, 2024, doi:10.3390/jof10010034_

Round 1
Reviewer 1 Report
Comments and Suggestions for Authors
The authors reviewed genetic maturation methods and applications of industrial microorganism, Aspergillus oryzae. Although the contents in the manuscript is interesting, many of subjects are the same as the previous review written by Jin et. al. (Front. Microbiol. 12:644404, 2021). Furthermore, in the point of view of an A. oryzae researcher, the descriptions of selectable markers, and promoters in this manuscript are not proper. Unfortunately, this manuscript is not suitable for publishing in “Journal of Fungi”.
[Comments]
Line 129-130,
The inducible promoters are not always safe to use in A. oryzae.
In case of very strong inducible promoters (such as PamyB) in A. oryzae, basal revel expression (non-induced condition) may affect the phenotype and intracellular localization of the target proteins.
Line 138-142
Description of PthiA is not proper.
PthiA is one kind of riboswitch which represses the expression of target gene in the present of thiamin (Vitamin B1). Thiamin induces concentration-dependent “reduction” of PthiA expression (It means PthiA is not induced by without thiamin condition). Therefore, in general PthiA does not used in Heterologous protein expression in A. oryzae. Instead, PthiA is often used in the characterization of the gene, disruption of which is lethal for A. oryzae.
Line 184-186,
Blemb and AosodhB are not generally used in A. oryzae researchers. Because, using of these promoters needs special drug-resistant hosts. On the other hand, ptrA maker can use non- pyrithiamine-resistant host (normal A. oryzae strains). ptrA maker has been widely used in the A. oryzae researches.
Table 1, PxlnA
As the PxlnA is a prokaryotic promoter, it is not proper to include table 1.
(PxlnA from L. acidophilus is only used in Cre-loxP recombination system)
Furthermore, in the case of A. oryzae, “PxlnA” indicates the promoter of endo-1,4-b-xylanase A gene (xlnA, AO090103000141).
Titles of table 1 and 2 were not properly indicate the contents of the tables.
Table 1, PxyrA
b-gucronic acid -> b-glucronidase
Table 1. Pmelo
Pmelo -> PmelO (“O”, large caption)
Table 1 PhyA
Hemolysin -> Hemolysin-like-protein
Table 1 PhylA
glucose endonuclease -> endo-1,4-glucanase
Table 1
Appliance -> Application??, Usage??
Line 28
A. oryzae 3.024 -> A. oryzae 3.042
Line 146
pgaA and pgaB -> pgaA and pgaB (both “p” should be in italic)
Table 2
3.024 D /pyrG -> 3.042 DpyrG
Line 159
IIIa, IIIb -> Region IIIa, IIIb (in PagdA)
Reference section, name of the journals of references,
Both abbreviation and full name were used in the reference section.
In general, abbreviation is recommended to use in the reference section.
Reference,
The no journal name was indicated in Ref. No.88.
Comments on the Quality of English LanguageIn a total, quality of English Language is not bad. However, the tiles of table 1 and 2 is not proper for scientific paper.
Author Response
Thank you for your suggestions, all answers are in the form of a file display

Reviewer 2 Report
Comments and Suggestions for Authors
The work is written in good scientific language: clearly, meaningfully. Numerous engineering manipulations that were used in working with Aspergillus oryzae are brought together. In particular, the review covered numerous tools, such as cloning methods, regulatory elements for gene expression, transformation methods, genome editing, and heterologous expression strategies. However, dividing the review into such sections somewhat conflicts with its title. Think about how it can be reformulated, since the examples you listed relate not only to genetic engineering and systems biology, but also to metabolic engineering, between which there are differences, according to many classifications, for example here (doi: 10.3389/fbioe.2019.00036).
The review appears coherent because it brings together numerous tools that were used in the engineering of such an important biotechnological object as A. oryzae. However, it does not contain a single specific example, expressed in numbers, of the industrial application of the technologies being introduced. For example, when BGC for the production of strobilurin (from basidiomycetes) was transferred to A. oryzae, one variant of recombinant strains of A. oryzae was able to produce strobilurin (DOI: 10.1038/s41467-018-06202-4). However, these products were negligible and heavily contaminated with impurities. Therefore, such work was of purely scientific interest; For example, during the work, with heterologous expression of strobilurin, recombinant A. oryzae synthesized several new, previously unknown for natural strains of basidiomycetes, secondary metabolites, which were variants of the products of the strobilurin biosynthesis pathway. However, these studies have not been scaled up to produce strobilurin, agriculturally important fungicide, on a commercial scale.
In the introduction, you present a series of studies on the production of enzymes and secondary metabolites through heterologous expression in A. oryzae. For example, work (https://doi.org/10.1080/09168451.2018.1527210) where heterologously overexpressed the egt-1 and -2 genes of Neurospora crassa in A. oryzae led to the production of ergothioneine (231.0 mg/kg of media, 20 times higher than the wild type). Is it a lot or a little? Are advances in genetic and metabolic engineering currently used to obtain industrial strains of A. oryzae? Or were all existing industrial strains of A. oryzae obtained using classical strain improvement?
In order to build a bridge between the ongoing research and its biotechnological significance, provide examples of the use of Synthetic biology tools for the industrial use of modified A. oryzae strains in the Conclusions and perspectives section or in any other sections of the review. Or note that there are currently no such examples to show the current status of implementation of genetic engineering technologies for A. oryzae.
In general, the work deserves publication after minor corrections taking into account comments.
Comments:
Line 48
Transformation-associated recombination. Write without capitalization, as is usual for this term, unless using an abbreviation, and if it is not at the beginning of a sentence
Lines 75-77
Reword this sentence because the current version gives the erroneous impression that introduction of an 8.5-kb fragment from a basidiomycete results in heterologous strobilurin synthesis in A. oryzae NSAR1. This 8.5-kb fragment encodes the PKS from strobilurin BGC (representing the stpks1 gene, isolated from cDNA). And to obtain strobilurin, other biosynthetic genes were also transferred to A. oryzae, such as str11 (PAL), str8 (NHI), str10 (CoA ligase), str9 (FDO), str2 (Met1), str3 (Met2). Many of these later pathway genes were also cloned using recombination in yeast (in the Gateway method design).
Author Response
Thank you for your suggestions, all answers in the form of word display.

Reviewer 3 Report
Comments and Suggestions for Authors
Review for
Synthetic biology tools for engineering Aspergillus oryzae
By Hui Yang 1, 2, Chaonan Song 1,2, Chengwei Liu 1,2 * and Pengchao Wang 1,2*
Aspergillus oryzae is attracting a lot of interest in research
2,881 documents found
In Scopus database, Aspergillus AND oryzae in paper titles
This reflects a long story with humans
For over a thousand years, Aspergillus oryzae has been used in traditional culinary, including food fermentation, brewing, and flavoring industries.
----------------------------------
In recent years, A. oryzae has been extensively used in deciphering the pathway of natural product synthesis and valued-added compound bioproduction. At the same time, it is also increasingly used in modern biotechnology industries such as the production of enzymes and recombinant proteins.
In this nice review, the advancements of A. oryzae synthetic biology tools including DNA assembly technologies, gene expression regulatory elements, and genome editing system are discussed.
+ Authors also address the challenges associated with heterologous expression of A. oryzae.
Examples
Biosynthesis of pleuromutilin congeners using an Aspergillus oryzae expression platform
Alberti, F., Khairudin, K., Davies, J.A., ...Foster, G.D., Bailey, A.M.
Chemical Science, 2023, 14(14), pp. 3826–3833
High-efficient production of mushroom polyketide compounds in a platform host Aspergillus oryzae
Han, H., Yu, C., Qi, J., ...Xia, X., Liu, C.
Microbial Cell Factories,
_________________________________
Recently, integration of heterologous pathways into A. 33 oryzae's genome through precise genetic manipulations has yielded a diverse array of 34 enzymes and secondary metabolites, exemplified by ergothioneine [5], Orsellinic acid 35 [6], chevalone E [7], abscisic acid [8], aphidicolin [9], etc.
Fully true
_____________________________
As a conclusion, very nice work, will be useful for scientists
Author Response
Response to Reviewer 3 Comments
Q1、Aspergillus oryzae is attracting a lot of interest in research
2,881 documents found
In Scopus database, Aspergillus AND oryzae in paper titles
This reflects a long story with humans
For over a thousand years, Aspergillus oryzae has been used in traditional culinary, including food fermentation, brewing, and flavoring industries.
In recent years, A. oryzae has been extensively used in deciphering the pathway of natural product synthesis and valued-added compound bioproduction. At the same time, it is also increasingly used in modern biotechnology industries such as the production of enzymes and recombinant proteins.
In this nice review, the advancements of A. oryzae synthetic biology tools including DNA assembly technologies, gene expression regulatory elements, and genome editing system are discussed.
Authors also address the challenges associated with heterologous expression of A. oryzae.
Examples
Biosynthesis of pleuromutilin congeners using an Aspergillus oryzae expression platform
Alberti, F., Khairudin, K., Davies, J.A., ...Foster, G.D., Bailey, A.M.
Chemical Science, 2023, 14(14), pp. 3826–3833
High-efficient production of mushroom polyketide compounds in a platform host Aspergillus oryzae
Han, H., Yu, C., Qi, J., ...Xia, X., Liu, C.
Microbial Cell Factories, 2023, 22(1), 60
Recently, integration of heterologous pathways into A. 33 oryzae's genome through precise genetic manipulations has yielded a diverse array of 34 enzymes and secondary metabolites, exemplified by ergothioneine [5], Orsellinic acid 35 [6], chevalone E [7], abscisic acid [8], aphidicolin [9], etc.
Fully true
A1、Thank you for your reading and suggestions.
Reviewer 4 Report
Comments and Suggestions for Authors
The manuscript is interesting and easy to follow. However, the recently published articles related to bioproduct production studies through A.oryzae should be discussed, and these examples should be given in a table. The current studies could improve the quality of the manuscript.
Author Response
Response to Reviewer4 Comments
Q1、The manuscript is interesting and easy to follow. However, the recently published articles related to bioproduct production studies through A.oryzae should be discussed, and these examples should be given in a table. The current studies could improve the quality of the manuscript.
A1、Thank you very much for your careful review. In this review, we focused on the A.oryzae gene manipulation production tools rather than a summary of A.oryzae secondary metabolites. The summary of A.oryzae secondary metabolites has been described in detail at doi.org / 10.1016 / j.biotechadv.2018.02.001.
Round 2
Reviewer 1 Report
Comments and Suggestions for Authors
The authors reviewed bioengineering techniques of Aspergillus oryzae such as DNA assembly and genome editing etc. I agree with the comments from the authors. However, this manuscript needs revise.
[Major comments]
Line 139
The promoter inhibits the expression of target genes in ……
-> The thiamine thiazole synthase promoter (PthiA) inhibits the expression of target genes in ……
(The name of the promoter, PthiA, should be indicated in this sentence.)
Line 132-134
However, the use of inducible promoters in A. oryzae is not always safe. In A. oryzae, if the promoter (such as PamyB) is highly inducible, non-inducible conditions may affect the phenotype and intracellular localization of the target protein.
In previous review, I described this sentence as just a comment to the authors. This sentence may cause confusion of the readers. Therefore, I recommend to delete this sentence from the manuscript.
[Minor comments]
Line 150
Pmelo -> PmelO
Table 1
Hemolysin-like-protein -> hemolysin-like-protein
Author Response

(The authors gave the same response as above.)

Reviewer 4 Report
Comments and Suggestions for Authors
There is no comment.
Author Response
Thanks.